# D³CNNs: Dual Denoiser Driven Convolutional Neural Networks for Mixed Noise Removal in Remotely Sensed Images

Zhenghua Huang [1,2,3], Zifan Zhu [3], Zhicheng Wang [3], Xi Li [1,3], Biyun Xu [3], Yaozong Zhang [3] and Hao Fang [4,*]

1   School of Information and Artificial Intelligence, Nanchang Institute of Science and Technology, Nanchang 330108, China
2   Artificial Intelligence School, Wuchang University of Technology, Wuhan 430223, China
3   School of Electrical and Information Engineering, Wuhan Institute of Technology, Wuhan 430205, China
4   School of Electronic and Information Engineering, Wuhan Donghu University, Wuhan 430212, China
*   Correspondence: fanghao@wdu.edu.cn

**Abstract:** Mixed (random and stripe) noise will cause serious degradation of optical remotely sensed image quality, making it hard to analyze their contents. In order to remove such noise, various inverse problems are usually constructed with different priors, which can be solved by either model-based optimization methods or discriminative learning methods. However, they have their own drawbacks, such as the former methods are flexible but are time-consuming for the pursuit of good performance; while the later methods are fast but are limited for extensive applications due to their specialized tasks. To fast obtain pleasing results with combination of their merits, in this paper, we propose a novel denoising strategy, namely, Dual Denoiser Driven Convolutional Neural Networks (D³CNNs), to remove both random and stripe noise. The D³CNNs includes the following two key parts: one is that two auxiliary variables respective for the denoised image and the stripe noise are introduced to reformulate the inverse problem as a constrained optimization problem, which can be iteratively solved by employing the alternating direction method of multipliers (ADMM). The other is that the U-shape network is used for the denoised auxiliary variable while the residual CNN (RCNN) for the stripe auxiliary variable. The subjectively and objectively comparable results of experiments on both synthetic and real-world remotely sensed images verify that the proposed method is effective and is even better than the state-of-the-arts.

**Keywords:** remotely sensed images; mixed noise removal; alternating direction method of multipliers; U-shape network; residual CNN

## 1. Introduction

Remotely sensed images (RSIs) are "photographs" of the Earth's surface, which can truly and vividly reflect the current situation of the distribution, the relationship, and the changes in the interaction of surface features, from which rich information including vegetation, soil moisture, water quality parameters, and surface and sea temperature can be obtained [1]. This Earth resource information can play an important role in the fields of agriculture, forestry [2], water conservancy, oceans, and ecological environment [3], which is beneficial for us to remotely investigate resources [4], monitor the environment [5], and analyze and predict disasters [6]. However, due to the impact of detector and photon effects, the obtained remotely sensed images may be degraded by both stripe noise (caused by the different responses of each detector) and random noise (which is mainly additive Gaussian white noise (AGWN) produced by photon effects) [7], which can be formulated as

$$y = x + s + n, \tag{1}$$

where $y$ is the observed image, $x$ is the latent clean image, $s$ is the stripe noise, and $n$ is the AGWN. It is an impossible task to simultaneously obtain $s$ and $x$ using Equation (1). To solve the problem, the common idea is the usage of a two-step method, such as first denoising [8–38] then destriping [39–55]. Such strategy may smooth many useful structures of image $x$. An example is shown in Figure 1, from which we can observe that many rich structures are smoothed. It is absolutely independent to reduce the two typical noise using the two-step method that may impair the structures of stripe or images and further smooth rich details. Therefore, one typical noise should be considered when the other is processed. To address this issue and preserve as many fine structures as possible, different regularizations on $x$ and $s$ are both considered to build a unified restoration model [56], which can be solved by the optimization methods (such as alternating direction method of multipliers (ADMM) and split Bregman). The procedures of these methods can be briefly presented as follows:

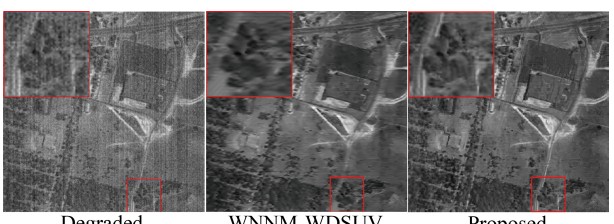

<center>Degraded　　　　　WNNM-WDSUV　　　　　Proposed</center>

**Figure 1.** A visual comparison example between a combination method and our unified scheme.

1. According to the Bayes' theorem, the estimation of $x$ and $s$ with the posterior distribution $P(x, s|y)$ can be converted into the following equation

$$P(x, s|y) \propto P(y|x, s)P(x)P(s),\qquad(2)$$

   where $P(y|x, s)$ is a likelihood prior which can be presented as

$$P(y|x, s) \propto exp\left\{-\frac{1}{2}\|y - x - s\|_2^2\right\},\qquad(3)$$

   where $\|\cdot\|_2^2$ is the $l_2$-norm function. $P(x)$ and $P(s)$ are respectively the prior probabilities of $x$ and $s$, which are used to obtain optimization solutions being closed to the actual values. With proper parameters $\lambda_1$ and $\lambda_2$, the prior probability $P(x)$ is written as

$$P(x) \propto exp\{-\lambda_1 \Phi(x)\},\qquad(4)$$

   while $P(s)$ is defined as

$$P(s) \propto exp\{-\lambda_2 \Psi(s)\},\qquad(5)$$

   where $\Phi(\cdot)$ and $\Psi(\cdot)$ are different regularizations on $x$ and $s$, respectively. By inserting Equations (3) to (5) into Equation (2), the posterior distribution $P(x, s|y)$ is equivalent to

$$P(x, s|y) \propto P(y|x, s)P(x)P(s) \propto exp\left\{-\left(\frac{1}{2}\|y - x - s\|_2^2 + \lambda_1 \Phi(x) + \lambda_2 \Psi(s)\right)\right\}.\quad(6)$$

2. With the usage of the logarithmic transformation, the optimization solution of Equation (6) is transferred from maximizing the posterior distribution to minimizing the energy function $\mathcal{L}(x, s) = -log\{P(x, s|y)\}$ which is

$$\mathcal{L}(x, s) = \frac{1}{2}\|y - x - s\|_2^2 + \lambda_1 \Phi(x) + \lambda_2 \Psi(s).\qquad(7)$$

3. The optimization solution of model (7) is solved with the employment of ADMM or split Bregman by introducing auxiliary variables.

## 2. Related Works

The stripe noise coexisting with the random noise makes it difficult to formulate the mixed noise with an explicit expression due to the nonindependent as well as nonidentical property. Usually, a unified energy model $\mathcal{L}$ is constructed by different penalty priors $\Phi$ and $\Psi$ respectively on image $x$ and stripe $s$ to separate them from image decomposition perspective [43], in which the stripe is equal to the image. According to the categories of priors, they are divided into the following three categories:

- *Sparsity-based priors:* These methods viewed that the image especially the stripe is sparse, so different priors, such as gradient-based variation, dictionary-based learning, and low-rank recovery, are combined to constrain the models for pursuing the optimal approximate solution. For instance, Huang et al. [56] proposed a uniform mixed noise removal model by the employment of joint analysis and weighted synthesis sparsity priors (JAWS). Chang et al. [57] employed unidirectional total variation and sparse representation (UTVSR) to simultaneously destripe and denoise remote sensing images. Xiong et al. [58] proposed a spectral-spatial $L_0$ gradient regularized low-rank tensor factorization method for hyperspectral denoising. Zheng et al. [59] removed mixed noise in hyperspectral images via low-fibered-rank regularization. Liu et al. [60] used the global and local sparsity constraints for a unified model construction to simultaneously estimation intensity bias and remove stripe noise in noisy infrared images. Zeng et al. [61] proposed a hyperspectral image restoration model with global $L_{1-2}$ spatial-spectral total variation regularized local low-rank tensor recovery. Xie et al. [62] denoised hyperspectral images via non-convex regularized low-rank and sparse matrix decomposition. Hu et al. [63] proposed a restoration method that can simultaneously remove Gaussian noise and stripes using adaptive anisotropy total variation and nuclear norms. Wang et al. [64] presented a $l_0 - l_1$ Hybrid total variation model for hyperspectral image mixed noise removal and compressed sensing. Wang et al. [65] exploited nonconvex logarithmic penalty for hyperspectral image denoising. These methods pursued their exciting denoising performance at the cost of expensively computational complexity.

- *Sparsity-based priors with joint of deep CNN denoiser prior:* Recently, deep convolutional neural network (CNN) as a prior for a specialized task has been popular applied in various fileds, especially in image restoration, due to its fast speed and large modeling capacity. Such property had been induced as an image prior to solve the inverse problem of image restoration [66–68], and had a considerable advantage. Inspired by its encouraging performance, Huang et al. [69] exploited deep CNN prior with the combination of unidirectional variation prior (UV-DCNN) to simultaneously destriping and denoising optical remotely sensed images. Zeng et al. [70] used CNN denoiser prior regularized low-rank tensor recovery for hyperspectral image restoration. These unfolding image denoising methods interpreted a truncated unfolding optimization as an end-to-end trainable deep network and thus usually produced pleasing results with fewer iterations using additional training for each task [68].

- *Discriminative learning prior:* As the Gaussian white noise and the stripe noise are both additive, so there are also various CNN-based denoising methods proposed to obtain both the image and the stripe. For example, He et al. [71] proposed a deep-learning approach to correct single-image-based nonuniformity in uncooled long-wave infrared detectors. Chang et al. [72] introduced a deep convolutional neural network (DCNN), named as HSI-DeNet, for HSIs' noise removal. Zhang et al. [73] employed a spatial-spectral gradient network to remove hybrid noise in hyperspectral image. Luo et al. [74] suggested a spatial–spectral constrained deep image prior (S2DIP), which simultaneously capitalize the high model representation ability brought by the CNN in an unsupervised manner and does not need any extra training data. Despite

the effectiveness of these methods, the CNN models are pretrained and cannot be jointly optimized with other parameters.

Inspired by the ideas in Refs. [66–69], in this paper, we proposed a unified mixed noise removal framework, named as Dual Denoiser Driven Convolutional Neural Networks (D³CNNs), to take advantages of both the optimization- and discriminative-learning-based methods. The flowchart of the proposed D³CNNs approach is shown in Figure 2. The main contributions of this paper are as follows:

1.  A unified mixed noise removal (MNR) framework, named as Dual Denoiser Driven Convolutional Neural Networks (D³CNNs), is proposed by using the CNN based denoiser and striper priors.
2.  Two deep denoiser/striper priors, respectively trained by a highly flexible U-shape denoiser and an effective residual learning strategy, are plugged as two modular parts into a half quadratic splitting based iterative algorithm to solve the inverse problem.
3.  Quantitative and qualitative results of experiments on both synthetic and real-world images validate the effectiveness of the proposed mixed noise removal scheme and even outperforms other advanced denoising approaches.

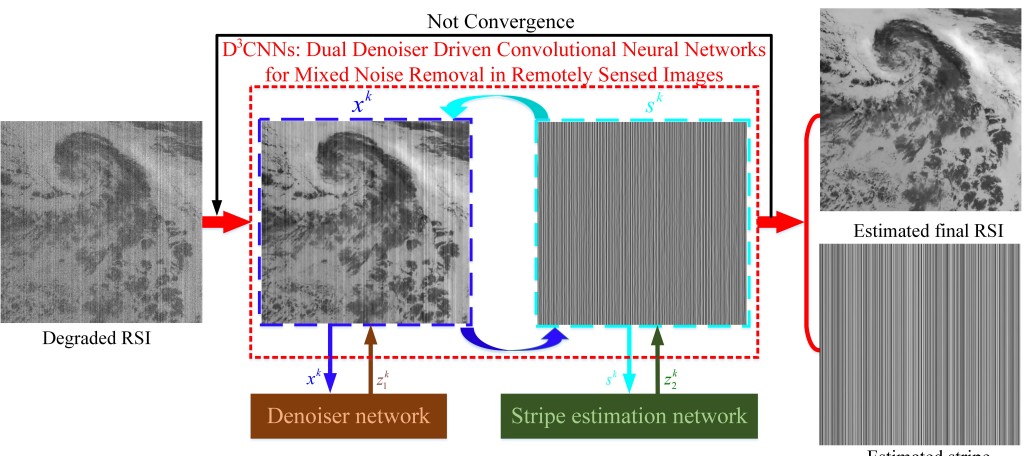

**Figure 2.** Flowchart of the proposed D³CNNs method.

## 3. Dual Denoiser Driven Convolutional Neural Networks

Although various variable splitting algorithms can be employed to solve model (7), in this paper, we adopt half quadratic splitting (HQS) method due to its simplicity and fast convergence [68].

### 3.1. Half Quadratic Splitting (HQS) Algorithm

In order to plug the denoiser prior as well as the striper prior into the optimization procedure of Equation (7), two auxiliary variables $z_1$ and $z_2$ are introduced in HQS to decouple the data term and prior terms of Equation (7) and to reformulate it as a constrained optimization problem given by

$$\mathcal{L}(x,s) = \frac{1}{2}\|y - x - s\|^2 + \lambda_1 \Phi(z_1) + \lambda_2 \Psi(z_2), s.t. \quad z_1 = x, \quad z_2 = s. \tag{8}$$

Then, Equation (8) is solved by minimizing the following cost function

$$\mathcal{L}_{\mu_1,\mu_2}(x,s) = \frac{1}{2}\|y - x - s\|^2 + \lambda_1 \Phi(z_1) + \frac{\mu_1}{2}\|x - z_1\|^2 + \lambda_2 \Psi(z_2) + \frac{\mu_2}{2}\|s - z_2\|^2, \tag{9}$$

where $\mu_1$ and $\mu_2$ are penalty parameters that vary iteratively in a non-descending order. The problem (9) can be addressed by the usage of the alternating direction method of

multipliers (ADMM), which iteratively solves the following subproblems for each variable while keeping the rest variables fixed:

$$\min_{x,s} \frac{1}{2}\|y - x - s\|^2 + \frac{\mu_1}{2}\|z_1 - x\|^2 + \frac{\mu_2}{2}\|z_2 - s\|^2, \tag{10}$$

$$\min_{z_1} \lambda_1 \Phi(z_1) + \frac{\mu_1}{2}\|z_1 - x\|^2, \tag{11}$$

$$\min_{z_2} \lambda_2 \Psi(z_2) + \frac{\mu_2}{2}\|z_2 - s\|^2. \tag{12}$$

The subproblem (10) can be further separated into two subproblems, and then Equation (9) can be iteratively solved by the following subproblems:

$$x^{k+1} = \arg\min_{x} \frac{1}{2}\left\|y - x - s^k\right\|^2 + \frac{\mu_1}{2}\left\|z_1^k - x\right\|^2, \tag{13}$$

$$s^{k+1} = \arg\min_{s} \frac{1}{2}\left\|y - x^{k+1} - s\right\|^2 + \frac{\mu_2}{2}\left\|z_2^k - s\right\|_2^2, \tag{14}$$

$$z_1^{k+1} = \arg\min_{z_1} \frac{1}{2\left(\sqrt{\lambda_1/\mu_1}\right)^2}\left\|z_1 - x^{k+1}\right\|^2 + \Phi(z_1), \tag{15}$$

$$z_2^{k+1} = \arg\min_{z_1} \frac{1}{2\left(\sqrt{\lambda_2/\mu_2}\right)^2}\left\|z_2 - s^{k+1}\right\|^2 + \Psi(z_2). \tag{16}$$

As we can see, the data term and regularization term are separated into four individual subproblems. To be specific, Equations (13) and (14) are both quadratic regularized least-squares problems which have fast closed-form solutions:

$$x^{k+1} = \frac{y + \mu_1 z_1^k - s^k}{1 + \mu_1}, \tag{17}$$

$$s^{k+1} = \frac{y + \mu_2 z_2^k - x^{k+1}}{1 + \mu_2}. \tag{18}$$

According to Bayesian perspective [75], the subproblem (15) corresponds to Gaussian denoising on the image $x^{k+1}$ by a Gaussian denoiser with noise level $\sqrt{\lambda_1/\mu_1}$ and the subproblem (16) corresponds to denoising the stripe image $s^{k+1}$ by a stripe restorer with noise level $\sqrt{\lambda_2/\mu_2}$. Consequently, the denoiser and the stripe restorer acted as two modular parts are plugged into the alternating iterations to solve Equation (8). To address this, Equations (15) and (16) can be rewritten as follows:

$$z_1^{k+1} = Denoiser(x^{k+1}, \sqrt{\lambda_1/\mu_1}), \tag{19}$$

$$z_2^{k+1} = striper(s^{k+1}, \sqrt{\lambda_2/\mu_2}). \tag{20}$$

From Equations (19) and (20), two benefits can be observed. First, the priors $\Phi(\cdot)$ and $\Psi(\cdot)$ can be implicitly replaced by a denoiser and a stripe restorer, respectively. Such a promising property can be jointly employed to solve many inverse problems, for instance, denoising and stripe restoration subproblems. Second, it is interesting to learn a DCNN denoiser and a DCNN stripe restorer to replace Equations (19) and (20), respectively, so as to utilize the advantages (such as high flexibility and efficiency as well as powerful modeling capacity) of DCNN.

### 3.2. U-Shape Denoiser Network

U-Net, known as an effective and efficient tool for image-to-image translation, fused multiscale features by concatenating the feature maps of the downsampling layers and the corresponding upsampling layers [68,76,77]. However, it may have two drawbacks. One

is that the information may be lost when using stride convolution operation. The other is that its modeling capacity is limited. To capture as much information for constructing the corrupted pixel as possible, the receptive field usually needs to be successively enlarged by the employment of convolution in CNN, which can be solved by increasing the filter size or the depth. At present, the existing popular way is to use $3 \times 3$ filter with a large depth. However, this method may cause a highly computational burden. Therefore, we replace traditional convolution (Conv) with dilate convolution (DConv) such that DConv can enlarge the receptive field while inheriting the superiorities of $3 \times 3$ Conv. For addressing the second issue, the residual network is employed due to its superior modeling capacity by stacking multiple residual blocks [68]. By introducing DConv and integrating residual learning modular (RLM) into U-Net, the proposed denoiser prior network, named as U-shape denoiser network (USD-Net), is modeled, the flowchart of which is shown in Figure 3.

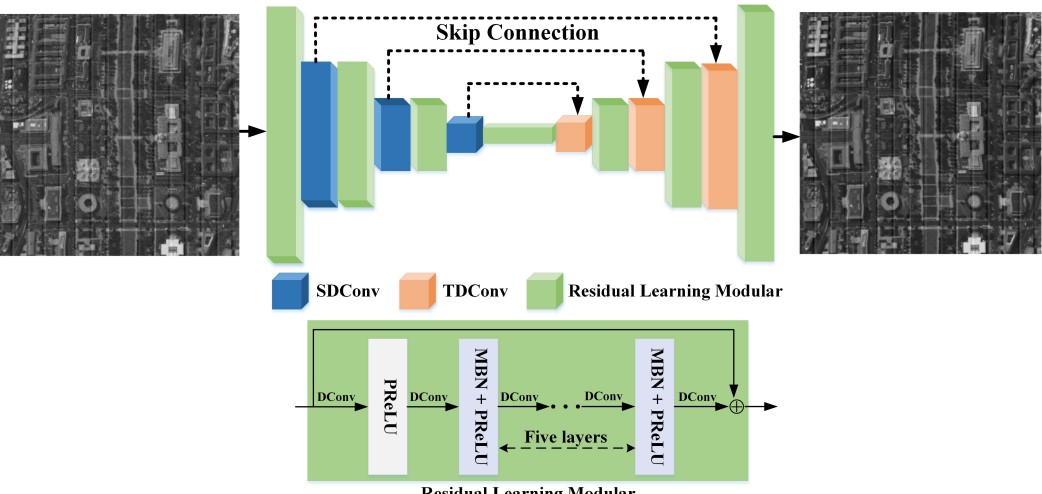

**Figure 3.** Flowchart of the proposed U-shape denoiser network. "SDConv" represents strided DConv while "TDConv" is transposed DConv, "⊕" is an adding operation.

Note that: (1) In RLM, the batch normalization (BN) and rectified linear unit (ReLU) are respectively replaced by the momentum batch normalization (MBN) [78] and parametric rectified linear unit (PReLU), as MBN can solve the underfitting problem caused by a small batch size of BN while PReLU is utilized for the nonlinearity to generate high quality estimation with less filters. The dilation factors from the first layer to the last layer are respectively set to 1, 2, 3, 4, 3, 2, and 1, which aggregates multiscale contextual information without losing resolution or increasing the depth of the network. The equivalent receptive field of each layer is 3, 5, 7, 9, 7, 5, and 3. In RLM of each scale layer, five successive "MBN + PReLU" blocks are adopted. (2) The USD-Net has four-scale layers, $2 \times 2$ strided dilated convolution (SDConv) is employed between each downscaling layer while $2 \times 2$ transposed dilated convolution (TDConv) is exploited between each upscaling layer. (3) The same scale between SDConv and TDConv has an identity skip connection. (4) The channels in each scale layer are gradually increased from 64 to 128 to 256 to 512.

### 3.3. Stripe Estimation Network

For stripe estimation, a deep residual convolutional network (SE-Net), which had been proved efficiently and effectively in various fileds [33,34,79], is employed. Similar to USD-Net, the components BN and ReLU of the traditional blocks are respectively replaced by the MBN and PReLU. The architecture of SE-Net is shown in Figure 4, and its depth is 16; such deeper layers can enlarge receptive field to obtain more contextual information for constructing the stripe image precisely. According to the differences of the components and the channels in the blocks, the whole layer of the SE-Net is decomposed into six sublayers,

for instance: (1) the first sublayer is the "Conv + PReLU" block including 64 filters with the size of $3 \times 3 \times 64$, each sublayer from the second to the six sublayers contains three blocks; (2) The sublayers from the second to the six layers, respectively named as CMP 1, CMP 2, CMP3, CMP 2, and CMP 1, are symmetrical according to the channels, each of which contains three components: Conv, MBN, and PReLU; (3) The channels from the first to the six sublayers are respectively 64, 64, 128, 256, 128, and 64 with dilation 1.

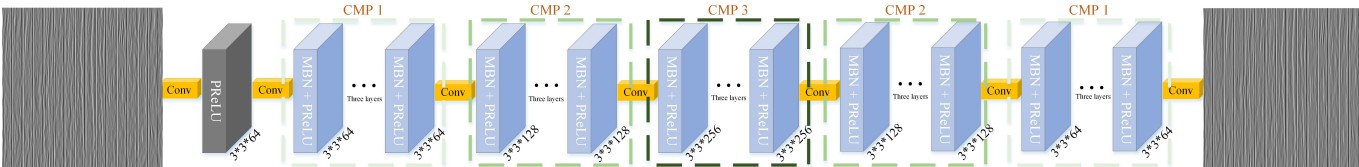

**Figure 4.** Flowwork of the proposed stripe estimation network.

*3.4. Loss Function*

As there are two networks respectively for different tasks, so they are pretrained with the usage of loss functions to guarantee a stable convergence for favorable results. For accurately estimating the clean image and the stripe image, the global and local information are both important. To this point, the most widely used loss function (mean squared error, MSE)

$$\mathcal{LF}_{MSE} = \frac{1}{N} \sum_{i=1}^{N} \|\Phi(x_i) - g_i\|_2^2 \tag{21}$$

is exploited as a global loss to perform global constraints, where $\Phi(\cdot)$ denotes USD-Net, $N$ represents the total number of training image pairs $\{x_i, g_i\}_{i=1}^{N}$, $x_i$ and $g_i$ respectively denote the $i$th latent clean image in the training database and its corresponding ground-truth image. Meanwhile, a local loss $\mathcal{LF}_G$ is formed by the $l_1$-norm on the gradient information for artifacts prevention and image structure preservation in the estimated result, which is formulated as

$$\mathcal{LF}_G = \frac{1}{N} \sum_{i=1}^{N} \|\nabla\Phi(x_i) - \nabla g_i\|_1, \tag{22}$$

where $\nabla$ denotes the difference in the horizontal and vertical directions. Finally, the whole loss function of USD-Net is defined as

$$\mathcal{LF}_{USD-Net} = \mathcal{LF}_{MSE} + \beta\mathcal{LF}_G \tag{23}$$

to be minimized for training the USD-Net $\Phi(\cdot)$, where $\beta$ is the weighting parameter to balance the effects of the two losses. Such loss function (23) is also used to train the SE-Net $\Psi(\cdot)$, where the USD-Net mapping function $\Phi$, the latent clean image $x$, and the ground-truth image $g$ in the global and local losses are respectively replaced by the SE-Net mapping function $\Psi$, the contaminated stripe $s$, and the ground-truth stripe $T$.

Taking all above procedures into account, we can conclude the optimization of the proposed dual denoiser driven convolutional neural networks for remotely sensed image restoration as shown in Algorithm 1.

---

**Algorithm 1** The Optimization of Dual Denoiser Driven Convolutional Neural Networks for Remotely Sensed Image Restoration

---

**Initial Setting:** Observed degraded image $y$, parameters $\lambda_1$ and $\lambda_2$, iteration number $K$, initial noise level $\sigma_1^0$ and $\sigma_2^0$, $z_1^0 = z_2^0 = 0$, and two pretrained networks (denoiser in Equation (19) and striper in Equation (20)).

**while** Convergence criterion Equations (24) and (25) or $k \leq K$ is not satisfied **do**

    1: Computing $x^k$ using Equation (17);

    2: Computing $s^k$ using Equation (18);

    3: Calculating $z_1^k$ using Equation (19);

    4: Calculating $z_2^k$ using Equation (20);

    5: Updating $k$: $k = k + 1$.

**end while**

**Output:** Latent clean image $x^k$ and stripe $s^k$.

---

## 4. Experimental Results and Discussion

### 4.1. Experimental Preparation

#### 4.1.1. Experimental Environment And Data

The denoiser as well as stripe estimation models are trained in MATLAB (R2015b) environment with MatConvNet package [80] on a PC with Intel Core i7-5960X CPU 3.0 GHz 16.0 GB memory associated with a Nvidia GTX 1080Ti GPU. The test data are downloaded from [81–83]. In [81], the MODIS aboard Terra and Aqua level 1B data contain 36 spectral bands, in which the data of the 36th band are degraded by both stripe and AGWN noise. Hyperspectral data in [82] and multispectral data in [83] are selected to test the structure preservation ability. The selected test images are cropped to 162,752 patches of size $35 \times 35$ and are separated randomly into training and test images with a ratio of 8:2.

#### 4.1.2. Experimental Parameters Setting

The denoiser learning network is trained by using stochastic gradient descent (SGD) [84] with a learning rate of $10^{-4}$ and is finished after 60 epochs while the stripe estimation network is trained by employing ADAM solver [85] with a learning rate of $10^{-5}$ and is finished after 300 epochs. The parameter $\beta$ in Equation (23) is empirically set to 0.5. From Equation (13) to Equation (16), we can find that there are five involved parameters including two regularization parameters $\lambda_1$ and $\lambda_2$, two penalty parameters $\mu_1$ and $\mu_2$, and iteration number $K$ to be set. Generally, the regularization parameters $\lambda_1$ and $\lambda_2$ come from the prior terms and keep fixed during iterations, and they are usually set as an empirical range for favorable performance, such as $\lambda_1 \in [0.21, 0.53]$ and $\lambda_2 \in [0.19, 0.57]$. In our experiments, we fix them to 0.25 and 0.21, respectively. Theoretically, the noise level ($\sigma_1 = \sqrt{\lambda_1/\mu_1}$ or $\sigma_2 = \sqrt{\lambda_2/\mu_2}$) is gradually decreasing during iterations, resulting in the continuous increment of the penalty parameters $\mu_1$ or $\mu_2$. In this paper, the initial $\sigma_1^0$ is fixed to 29 and the final $\sigma_1^K$ is determined by the image noise level (which is usually less than 29) while the initial $\sigma_2^0$ is set to 49 and the final $\sigma_2^K$ is determined by the stripe's intensity. Both $\sigma_1$ and $\sigma_2$ are uniformly sampled from the initial noise level to the final one in log space. The convergence criterion is as follows:

$$\frac{\left\| x^{k+1} - x^k \right\|_2^2}{\left\| x^k \right\|_2^2} < \eta_1 \tag{24}$$

and

$$\frac{\left\| s^{k+1} - s^k \right\|_2^2}{\left\| s^k \right\|_2^2} < \eta_2, \tag{25}$$

where $\eta_1 = 3 \times 10^{-4}$ and $\eta_2 = 2 \times 10^{-4}$ in the following experiments. When the convergence criterion is not satisfied, the total iteration number $K$ is set to 29, which is large enough to get a excited performance.

### 4.1.3. Compared Methods and Evaluation Indexes Selection

To verify the efficiency of the proposed D³CNNs, several state-of-the-arts derived from different categories are selected to be compared, including two-stage mixed noise removal (first denoising (Weighted Nuclear Norm Minimization, WNNM [26]), then destriping (Weighted Double-Sparsity Unidirectional Variation, WDSUV [49]) (WNNM-WDSUV), model-based methods (UTVSR [57] and JAWS [56]), semi-discrimination learning method (UV-DCNN [69]), and full-discrimination learning method (HSI-DeNet [72]). In the synthetic experiments, the synthesized RSIs are noised by AGWN ranged from 15 to 30 with a step size of 5 and stripe with three intensities (10, 30, and 50) as well as three proportions (0.1, 0.4, and 0.6), which are similar to those in Ref. [43]. Except for the visual comparisons, the objective indexes are also selected to quantitatively assess their ability of AGWN and stripe noise removal. For instance, two referenced indices, peak signal-to-noise ratio (PSNR) and structural similarity index (SSIM) [34], are employed to respectively assess the capability of noise reduction and structure preservation in synthetic experiments. While four reference-free metrics, mean of the mean relative deviation (MMRD) to evaluate the performance in retaining fine details of noise-free sharp regions [49], Q-Metric (QM) to evaluate the denoising performance [59], mean of the inverse coefficient of variation (MICV) to reflect the level of the remaining stripe noise in homogeneous regions [86], and natural image quality evaluator (NIQE) to evaluate the quality of the improved results [87], are employed for quantitative evaluation in the real-world experiments. In these indices, small MMRD and NIQE values depict that the estimated image quality is quite encouraged while large values of other metrics indicate that the results are pleasing.

In the following test experiments, eight synthetic RSIs (as shown in Figure 5) and seven real-life degraded RSIs (as shown in Figure 6) are selected to be experimented for the verification of the efficiency of the proposed D³CNNs strategy.

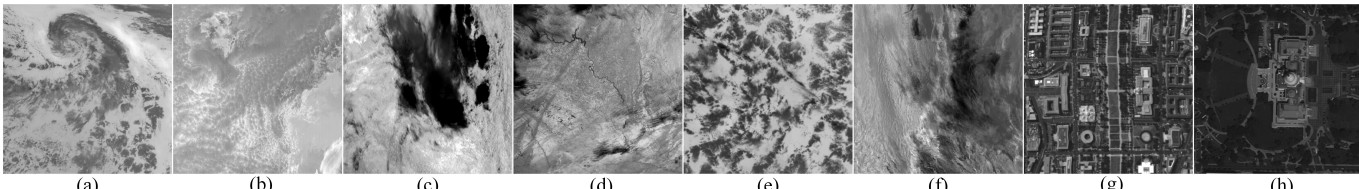

(a)    (b)    (c)    (d)    (e)    (f)    (g)    (h)

**Figure 5.** Eight testing images respectively from Terra MODIS data ((**a**–**c**) named as *STM1*, *STM2*, and *STM3*), Aqua MODIS data ((**d**–**f**) named as *SAM1*, *SAM2*, and *SAM3*), Hyperspectral data ((**g**) Washington DC Mall, *SWDCM*), and Multispectral data ((**h**) Washington DC, *SWDC*).

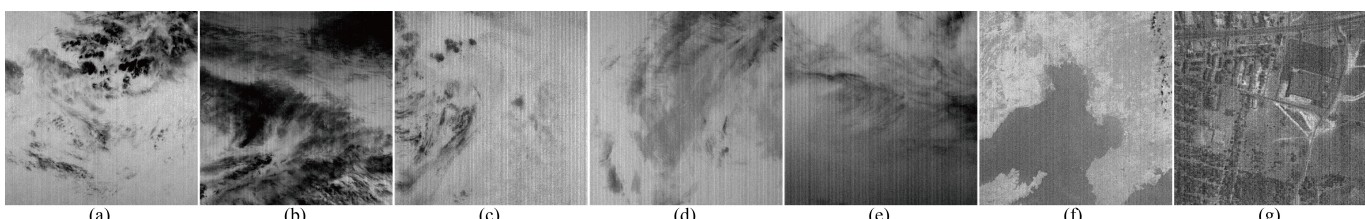

(a)    (b)    (c)    (d)    (e)    (f)    (g)

**Figure 6.** Eight real-life images respectively from Aqua MODIS data ((**a**–**c**) named as *RAM1*, *RAM2*, and *RAM3*), Terra MODIS data ((**d**–**f**) named as *RTM1*, *RTM2*, and *RTM3*), and Hyperspectral ((**g**) *urban*) data.

### 4.2. Discussion of Intermediate Results

Figures 7c–e and 8h–j respectively provide the visual results of $x^k$ and $s^k$ at different iterations on the testing images from Figure 5, while both Figures 7f and 8f show the PSNR

convergence curves for $x^k$ and $s^k$. From the figures, several observations can be concluded as follows: First, the deep denoiser and the deep striper priors play the important roles of noise removal and stripe estimation, leading to a noise-free image $x$ and a clean stripe $s$. Second, compared with intermediate results, the final results including $x$ and $s$ contain more fine details while they are more visually similar to the ground-truths, meaning that Equations (13) and (14) can iteratively recover the details with the help of two deep priors. Third, according to Figures 7f and 8f, $x^k$ and $s^k$ enjoy a fast convergence to a fixed point.

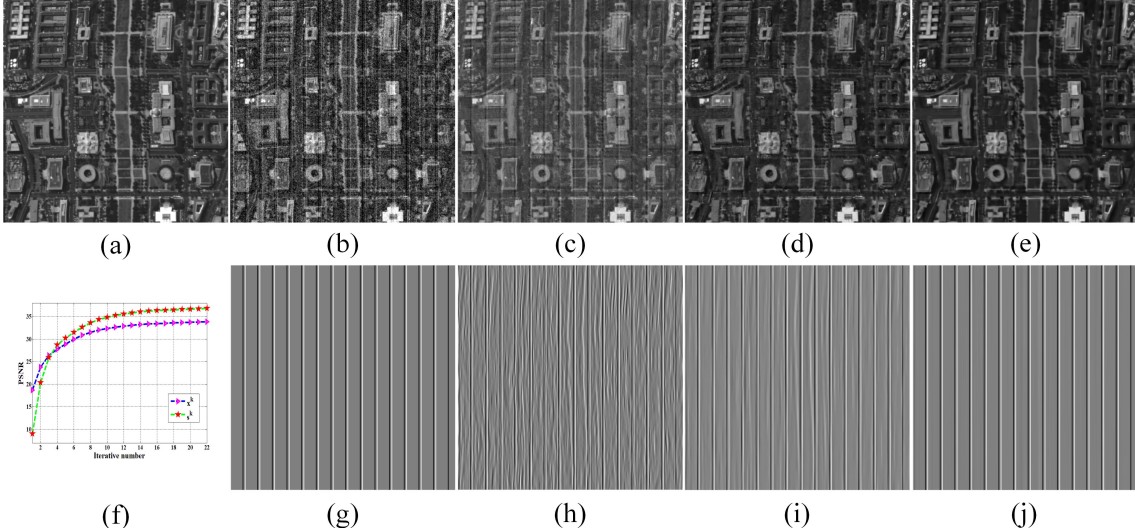

**Figure 7.** The intermediate results of $x^k$ and $s^k$ on the *SWDCM* image in Figure 5g at different iterations. Where noise level $\sigma = 25$, the proportion $P$ and the intensity $I$ of the **periodical** stripe are respectively 0.1 and 50, and the width of each stripe is 20 pixels. (**a**) Ground-truth. (**b**) Degraded *SWDCM* image. (**c**–**e**) Visual images respectively at the 5rd, 13th, and 19th. (**f**) Convergence curves of $x$ and $s$. (**g**) Ground-truth stripe. (**h**–**j**) Visual stripes respectively at the 5th, 13th, and 19th iteration.

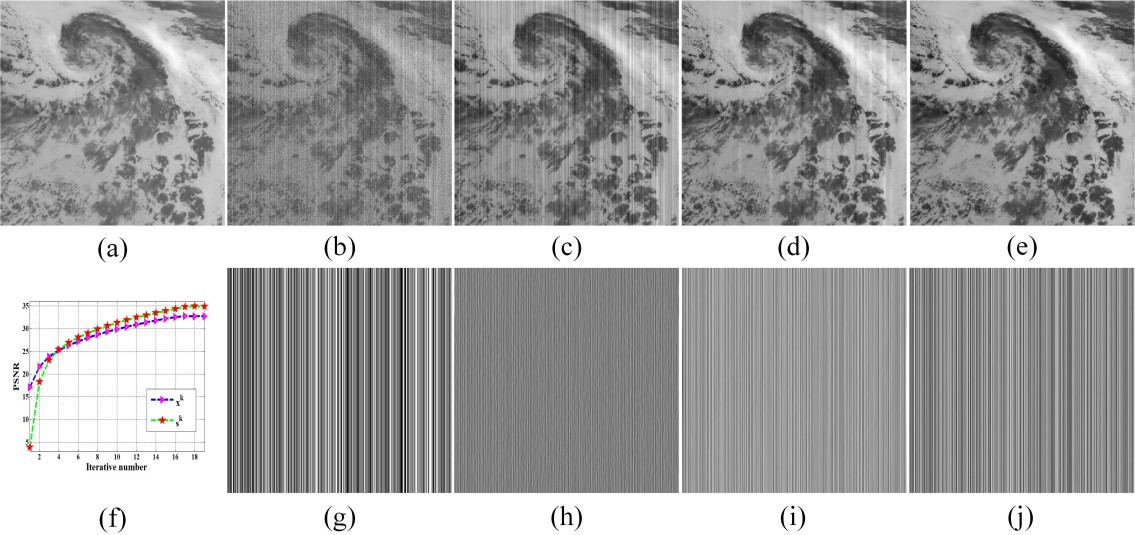

**Figure 8.** The intermediate results of $x^k$ and $s^k$ on the *STM1* image in Figure 5a at different iterations. Where noise level $\sigma = 30$, the proportion $P$ and the intensity $I$ of the **non-periodical** stripe are respectively 0.4 and 30. (**a**) Ground-truth. (**b**) Degraded *STM1* image. (**c**–**e**) Visual images respectively at the 3rd, 11th, and 21th. (**f**) Convergence curves of $x$ and $s$. (**g**) Ground-truth stripe. (**h**–**j**) Visual stripes respectively at the 3rd, 11th, and 21th iteration.

*4.3. Experiments on Synthetic Rsis*

4.3.1. Qualitative Evaluation

To subjectively assess the efficiency of the D³CNNs approach, we select the visual comparisons of five synthetic RSIs degraded by AGWN with different noise level and the stripe with different types (including period, proportion, and intensity), as shown in Figures 9–12. From the figures, we can see that all of state-of-the-arts have great ability in denoising and removing stripe. However, their performance can be discriminated from the enlarged visual areas, of which the observations can be concluded as follows: First, many fine details, especially the structures along the stripe, of the image results yielded by the UTVSR (Figures 9a, 10a and 11) and WNNM-WDSUV (Figures 9b, 10b and 11) methods are over-smoothed. Second, the HSI-DeNet method estimates latent clean images from the view of image decomposition and preserves more details than the prior two approaches but is still subject to losing image details in stripe maps, resulting in the lost of many rich details, as shown in Figures 9c, 10c, 11 and 12. Third, the UV-DCNN method plugs the DCNN denoiser prior into the UV model, which reduces the interference of AGWN on the recovery of the stripe to a great extent. As shown in Figures 9e, 10e, 11 and 12, the image details in stripe maps become less while the details in estimated images are richer. Fourth, with deep feature analysis of image and stripe, the JAWS method uses the characteristics of both image and stripe as constrained priors for the construction of a unified model to restore image and stripe. Compared with the forward four method, the JAWS method produces the suitable visual results with more abundant details and more comfortable stripes, as shown in Figures 9f, 10f, 11 and 12. Finally, the proposed D³CNNs method yields better promising image and stripe results (as shown in Figures 9g, 10g, 11 and 12) on details preservation in image and stripe regularity, illustrating that our method is better than others in image restoration.

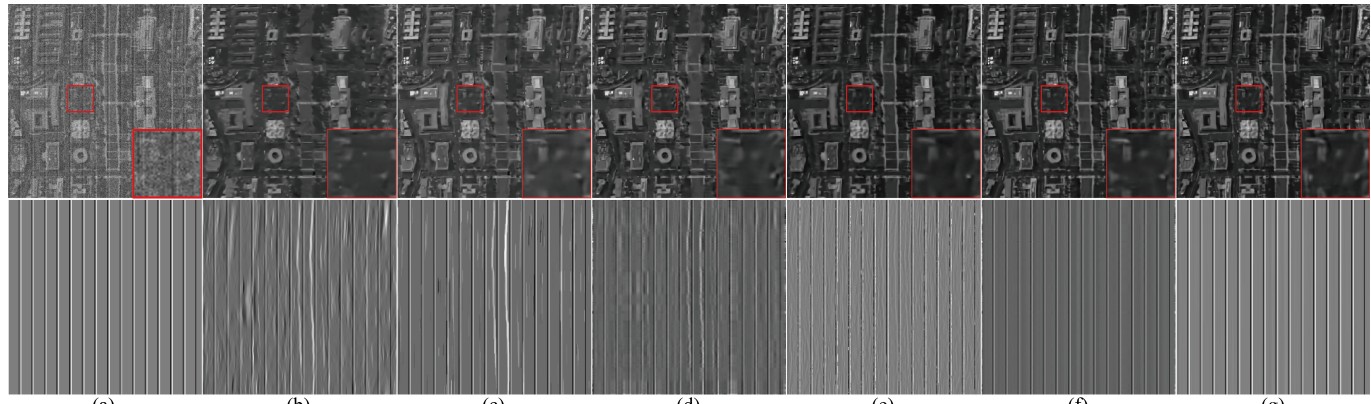

**Figure 9.** Visual comparisons on the *SWDCM* image in Figure 5g. Where noise level $\sigma = 25$, the proportion *P* and the intensity *I* of the **periodical** stripe are respectively 0.1 and 50. (**a**) Upper: Degraded *SWDCM* image. Down: Ground-truth stripe. (**b**–**g**) Estimated images and stripes respectively produced by the UTVSR, WNNM-WDSUV, HSI-DeNet, UV-DCNN, JAWS, and proposed methods.

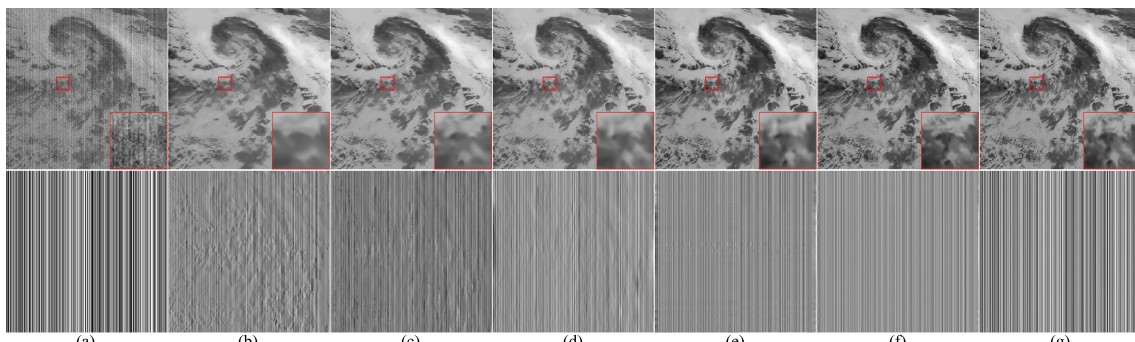

(a)　　　　(b)　　　　(c)　　　　(d)　　　　(e)　　　　(f)　　　　(g)

**Figure 10.** Visual comparisons on the *STM1* image in Figure 5a. Where noise level $\sigma = 30$, the proportion $P$ and the intensity $I$ of the **non–periodical** stripe are respectively 0.4 and 30. (**a**) Upper: Degraded *STM1* image. Down: Ground-truth stripe. (**b–g**) Estimated images and stripes respectively produced by the UTVSR, WNNM-WDSUV, HSI-DeNet, UV-DCNN, JAWS, and proposed methods.

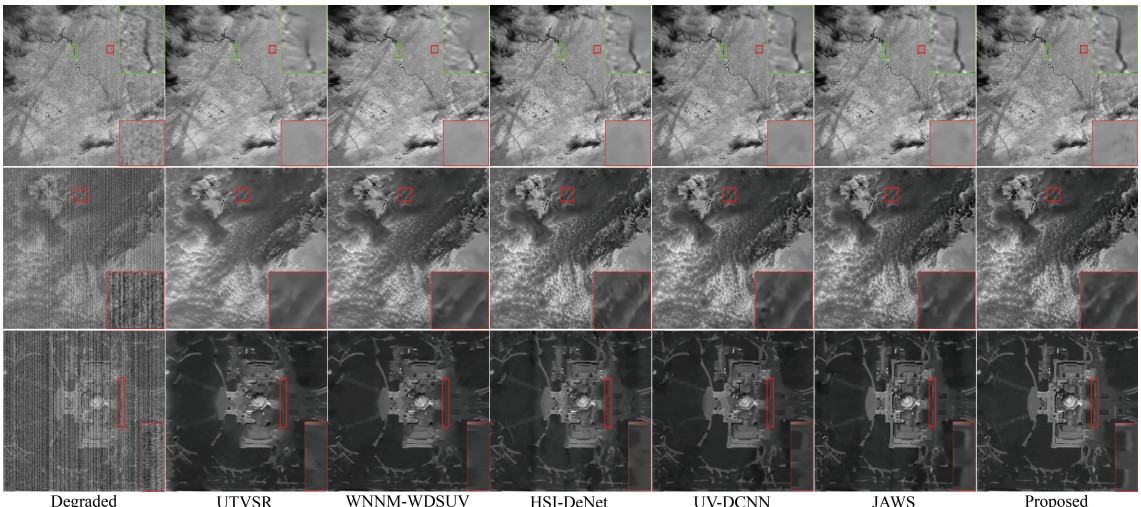

Degraded　　　UTVSR　　　WNNM-WDSUV　　　HSI-DeNet　　　UV-DCNN　　　JAWS　　　Proposed

**Figure 11.** Visual comparisons of the estimated results on the synthetic *SAM1*, *SWDC*, and *SWDC* images. For *SAM1*: $\sigma = 15$, $P = 0.4$, and $I = 10$. For *STM2*: $\sigma = 20$, $P = 0.4$, and $I = 30$. For *SWDC*: $\sigma = 25$, $P = 0.6$, and $I = 30$. The stripes on *SAM1* and *SWDC* are nonperiodical while the stripe on *STM2* is periodical.

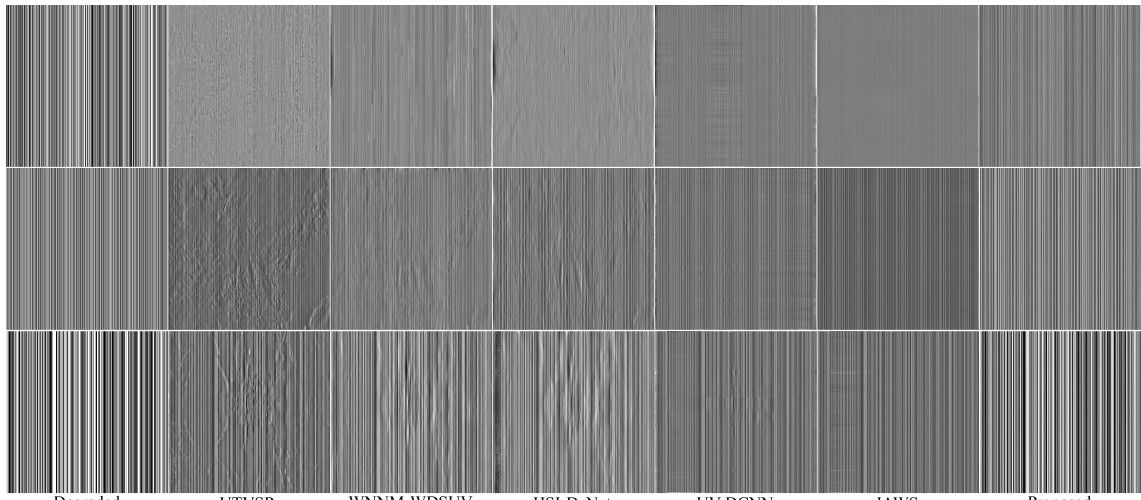

Degraded　　　UTVSR　　　WNNM-WDSUV　　　HSI-DeNet　　　UV-DCNN　　　JAWS　　　Proposed

**Figure 12.** Visual comparisons of the estimated stripes corresponding to the *SAM1*, *STM2*, and *SWDC* images in Figure 11.

4.3.2. Quantitative Assessment

As noise levels especially the stripe's types are different, we employ mean PSNR (MPSNR) and mean SSIM (MSSIM) to objectively evaluate each method. Such indices are defined as follows:

$$MPSNR = \frac{1}{N} \sum_{i_1=1}^{2} \sum_{i_2=1}^{3} \sum_{i_3=1}^{3} PSNR(i_1, i_2, i_3), \tag{26}$$

and

$$MSSIM = \frac{1}{N} \sum_{i_1=1}^{2} \sum_{i_2=1}^{3} \sum_{i_3=1}^{3} SSIM(i_1, i_2, i_3), \tag{27}$$

for each image at one noise level, where $N = 2 \times 3 \times 3 = 18$, $i_1 = 1$ represents the periodical stripe while $i_1 = 2$ is the non-periodical stripe. $i_2 = 1, 2$, and 3 respectively denote the stripe's intensity as 10, 30, and 50 while $i_3 = 1, 2$, and 3 respectively represent the stripe's proportion as 0.1, 0.4, and 0.6. Quantitative MPSNR and MSSIM results are respectively compared in Tables 1 and 2, from which the following two conclusions are involved: (1) for each image, both MPSNR and MSSIM are decreasing along with the increment of noise level, the larger noise level will seriously contaminate the images, making it more difficulty in the estimated results being closer to ground-truth. (2) The proposed D$^3$CNNs approach generates the highest MPSNR and MSSIM values on each image at the same noise level, illustrating that it is effective and even better than the state-of-the-arts, which is consistent with visual comparisons.

Run time, an important index to assess the efficiency of algorithms, is tested on images with different sizes, the results of which are shown in Table 3. From the table, we can find several observations: First, the run time of the discriminative learning methods (the results are marked with color font), even in the CPU version, is faster than that of the model-based optimization approaches. Second, the pure discriminative learning (HSI-DeNet, its results are marked with red font) gets the fastest speed on either CPU or GPU version under the same image size, which is reasonable and is used to learn a specialized prior (such as denoiser prior in UV-DCNN, the results are marked with green font) to be plugged into the model-based optimization methods for improving the computation time and boosting the modeling ability. Third, with the help of two deep priors, the proposed D$^3$CNNs generates the second fastest runtime, which is a little slower than that of HSI-DeNet, but it has much better MPSNR (as shown in Table 1) and MSSIM (as shown in Table 2) than HSI-DeNet.

According to the comprehensive consideration of the comparable results, we can get that D$^3$CNNs is a flexible and faithful method for image restoration.

**Table 1.** Quantitative MPSNR comparisons of the state-of-the-arts on the synthetic optical RSIs shown in Figure 5.

| Methods | STM1 | | | | STM2 | | | | STM3 | | | |
|---|---|---|---|---|---|---|---|---|---|---|---|---|
| | $\sigma = 15$ | $\sigma = 20$ | $\sigma = 25$ | $\sigma = 30$ | $\sigma = 15$ | $\sigma = 20$ | $\sigma = 25$ | $\sigma = 30$ | $\sigma = 15$ | $\sigma = 20$ | $\sigma = 25$ | $\sigma = 30$ |
| UTVSR | 28.14 | 26.86 | 25.9 | 25.22 | 29.96 | 28.83 | 27.58 | 27.1 | 33.12 | 32.04 | 31.05 | 30.25 |
| WNNM-WDSUV | 28.82 | 27.52 | 26.62 | 25.94 | 30.91 | 29.63 | 28.75 | 28.09 | 34.55 | 33.42 | 32.61 | 31.98 |
| HSI-DeNet | 29.02 | 27.71 | 26.77 | 26.04 | 31.02 | 29.72 | 28.78 | 28.1 | 34.71 | 33.56 | 32.72 | 32.08 |
| UV-DCNN | 29.05 | 27.77 | 26.89 | 26.24 | 31.13 | 29.87 | 29 | 28.34 | 34.78 | 33.6 | 32.84 | 32.11 |
| JAWS | 29.24 | 27.86 | 26.93 | 26.25 | 31.21 | 29.93 | 29.03 | 28.36 | 34.86 | 33.71 | 32.93 | 32.28 |
| Proposed | **29.68** | **28.15** | **27.46** | **26.76** | **31.59** | **30.47** | **29.73** | **28.69** | **35.18** | **34.06** | **33.38** | **32.79** |

**Table 1.** *Cont.*

| Methods | SAM1 | | | | SAM2 | | | | SAM3 | | | |
|---|---|---|---|---|---|---|---|---|---|---|---|---|
| | $\sigma=15$ | $\sigma=20$ | $\sigma=25$ | $\sigma=30$ | $\sigma=15$ | $\sigma=20$ | $\sigma=25$ | $\sigma=30$ | $\sigma=15$ | $\sigma=20$ | $\sigma=25$ | $\sigma=30$ |
| UTVSR | 28.67 | 27.14 | 25.98 | 25.21 | 28.46 | 27.13 | 26.02 | 25.29 | 28.12 | 26.92 | 26.11 | 25.27 |
| WNNM-WDSUV | 29.31 | 27.75 | 26.62 | 25.74 | 29.25 | 27.81 | 26.84 | 26.09 | 29.01 | 27.74 | 26.88 | 26.25 |
| HSI-DeNet | 29.24 | 27.77 | 26.68 | 25.82 | 29.36 | 27.98 | 26.95 | 26.14 | 29.17 | 27.88 | 26.98 | 26.29 |
| UV-DCNN | 29.47 | 28.01 | 26.89 | 26 | 29.45 | 28.11 | 27.14 | 26.4 | 29.31 | 27.97 | 27.13 | 26.5 |
| JAWS | 29.52 | 27.99 | 26.82 | 25.97 | 29.58 | 28.18 | 27.2 | 26.44 | 29.32 | 28.05 | 27.17 | 26.52 |
| Proposed | **29.75** | **28.64** | **27.33** | **26.47** | **30.02** | **28.83** | **27.91** | **27.08** | **29.84** | **28.68** | **27.79** | **27.01** |

| Methods | SWDCM | | | | SWDCM | | | |
|---|---|---|---|---|---|---|---|---|
| | $\sigma=15$ | $\sigma=20$ | $\sigma=25$ | $\sigma=30$ | $\sigma=15$ | $\sigma=20$ | $\sigma=25$ | $\sigma=30$ |
| UTVSR | 28.53 | 27 | 25.68 | 24.8 | 31.03 | 29.71 | 28.85 | 28.33 |
| WNNM-WDSUV | 29.44 | 27.76 | 26.51 | 25.53 | 32.41 | 30.96 | 29.87 | 29.03 |
| HSI-DeNet | 29.5 | 27.91 | 26.72 | 25.75 | 32.61 | 31.2 | 30.06 | 29.16 |
| UV-DCNN | 29.52 | 27.87 | 26.57 | 25.69 | 32.69 | 31.27 | 30.26 | 29.27 |
| JAWS | 29.62 | 28.02 | 26.77 | 25.84 | 32.83 | 31.41 | 30.33 | 29.48 |
| Proposed | **29.84** | **28.33** | **27.06** | **26.37** | **33.17** | **31.76** | **30.8** | **29.97** |

**Table 2.** Quantitative MSSIM comparisons of the state-of-the-arts on the synthetic optical RSIs shown in Figure 5.

| Methods | STM1 | | | | STM2 | | | | STM3 | | | |
|---|---|---|---|---|---|---|---|---|---|---|---|---|
| | $\sigma=15$ | $\sigma=20$ | $\sigma=25$ | $\sigma=30$ | $\sigma=15$ | $\sigma=20$ | $\sigma=25$ | $\sigma=30$ | $\sigma=15$ | $\sigma=20$ | $\sigma=25$ | $\sigma=30$ |
| UTVSR | 0.7872 | 0.7251 | 0.6626 | 0.6205 | 0.815 | 0.7592 | 0.7118 | 0.6825 | 0.8766 | 0.8587 | 0.8482 | 0.8342 |
| WNNM-WDSUV | 0.8196 | 0.7748 | 0.7209 | 0.6842 | 0.8237 | 0.7734 | 0.7476 | 0.7149 | 0.8821 | 0.872 | 0.8567 | 0.8413 |
| HSI-DeNet | 0.8216 | 0.7843 | 0.7257 | 0.6861 | 0.832 | 0.7771 | 0.7471 | 0.7157 | 0.8826 | 0.8715 | 0.8559 | 0.8424 |
| UV-DCNN | 0.8466 | 0.7944 | 0.7318 | 0.6898 | 0.8433 | 0.782 | 0.7482 | 0.7179 | 0.8895 | 0.8728 | 0.8577 | 0.8426 |
| JAWS | 0.8472 | 0.8003 | 0.7356 | 0.6917 | 0.8482 | 0.7867 | 0.7488 | 0.7187 | 0.8919 | 0.8724 | 0.8585 | 0.8462 |
| Proposed | **0.8518** | **0.8126** | **0.7415** | **0.7172** | **0.8527** | **0.8037** | **0.7524** | **0.7318** | **0.9011** | **0.8829** | **0.8617** | **0.8534** |

| Methods | SAM1 | | | | SAM2 | | | | SAM3 | | | |
|---|---|---|---|---|---|---|---|---|---|---|---|---|
| | $\sigma=15$ | $\sigma=20$ | $\sigma=25$ | $\sigma=30$ | $\sigma=15$ | $\sigma=20$ | $\sigma=25$ | $\sigma=30$ | $\sigma=15$ | $\sigma=20$ | $\sigma=25$ | $\sigma=30$ |
| UTVSR | 0.8908 | 0.8524 | 0.8216 | 0.7913 | 0.805 | 0.7249 | 0.6452 | 0.5886 | 0.748 | 0.6772 | 0.5981 | 0.5562 |
| WNNM-WDSUV | 0.8998 | 0.8662 | 0.8312 | 0.806 | 0.8127 | 0.7615 | 0.7094 | 0.6753 | 0.7846 | 0.7179 | 0.6661 | 0.6229 |
| HSI-DeNet | 0.905 | 0.8646 | 0.8335 | 0.8054 | 0.8225 | 0.7648 | 0.7137 | 0.6783 | 0.7763 | 0.7178 | 0.6673 | 0.6243 |
| UV-DCNN | 0.8996 | 0.8676 | 0.8309 | 0.8096 | 0.8266 | 0.7737 | 0.7168 | 0.6802 | 0.794 | 0.7196 | 0.6718 | 0.6277 |
| JAWS | 0.9064 | 0.8698 | 0.8334 | 0.8073 | 0.8272 | 0.7755 | 0.7155 | 0.6793 | 0.7992 | 0.7192 | 0.6728 | 0.6283 |
| Proposed | **0.9172** | **0.8813** | **0.8594** | **0.8216** | **0.8353** | **0.7962** | **0.7367** | **0.7015** | **0.8127** | **0.7533** | **0.7119** | **0.6527** |

| Methods | SWDCM | | | | SWDCM | | | |
|---|---|---|---|---|---|---|---|---|
| | $\sigma=15$ | $\sigma=20$ | $\sigma=25$ | $\sigma=30$ | $\sigma=15$ | $\sigma=20$ | $\sigma=25$ | $\sigma=30$ |
| UTVSR | 0.8912 | 0.8241 | 0.7802 | 0.7384 | 0.861 | 0.8287 | 0.7807 | 0.7477 |
| WNNM-WDSUV | 0.8937 | 0.8486 | 0.8094 | 0.7808 | 0.8645 | 0.83 | 0.7971 | 0.7725 |
| HSI-DeNet | 0.9003 | 0.8525 | 0.8123 | 0.7832 | 0.8619 | 0.8319 | 0.7976 | 0.7743 |
| UV-DCNN | 0.8978 | 0.8539 | 0.8139 | 0.7858 | 0.8651 | 0.8468 | 0.8066 | 0.7805 |
| JAWS | 0.9014 | 0.8549 | 0.8145 | 0.7833 | 0.8647 | 0.8476 | 0.8079 | 0.7817 |
| Proposed | **0.9153** | **0.8792** | **0.8386** | **0.8124** | **0.8878** | **0.8629** | **0.8237** | **0.8019** |

**Table 3.** Average computation time (Unit: Seconds) comparison on different image size. For learning-based methods, the computation times on CPU/GPU are both presented.

| Image Size | Methods | | | | | |
|---|---|---|---|---|---|---|
| | UTVSR | WNNM-WDSUV | HSI-DeNet | UV-DCNN | JAWS | Proposed |
| $256 \times 256$ | 653.923 | 108.714 | 1.048/0.016 | 1.077/0.035 | 874.675 | 1.068/0.024 |
| $512 \times 512$ | 2674.641 | 440.283 | 5.869/0.027 | 7.953/0.142 | 2937.424 | 6.667/0.073 |

### 4.4. Applications to the Real-World Degraded Rsis

To further verify the efficiency of the proposed D$^3$CNNs scheme, we apply it to the real-world degraded RSIs shown in Figure 6 and compare it to state-of-the-arts qualitatively and quantitatively. The visual comparisons of the estimated images and the calculated stripe maps are respectively shown in Figures 13 and 14, from which we can find that the details of images produced by our method are richer while the stripes are cleaner and more regular than those produced by others. Such conclusions denote that the results decomposed by our method are more faithful and are closer to original clean maps including images and stripes.

Table 4 shows the comparisons of quantitative results generated by the state-of-the-arts, from which several observations can be concluded: First, all of them yield considerable QM values, illustrating that they perform well on denoising. Our method produces the highest QM value for each RSI, showing the strongest ability of rich detail preservation. Then, the small difference of MICV values on the same RSI represents that they are all well in stripe removal in homogeneous regions. Meanwhile, for the same RSI, our method produces the smallest MMRD value, reflecting that it can generate pleasing results with more fine details in noise-free sharp regions. Finally, the smallest NIQE value obtained by our method on the same RSI demonstrates that it can better improve the image quality than others.

In sum, the comparisons of quantitative associated with qualitative results contribute to validate the effectiveness of the proposed D$^3$CNNs method.

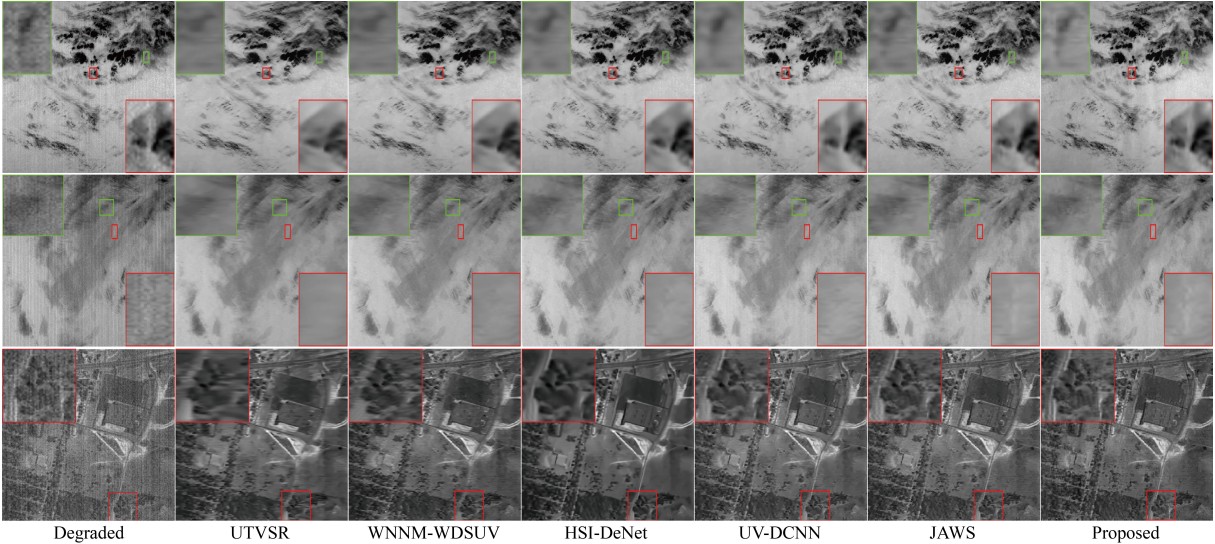

<div align="center">Degraded     UTVSR     WNNM-WDSUV     HSI-DeNet     UV-DCNN     JAWS     Proposed</div>

**Figure 13.** Visual comparisons of the estimated results on the real-world degraded *RAM1*, *RTM1*, and Hyperspectral *urban* images in Figure 6.

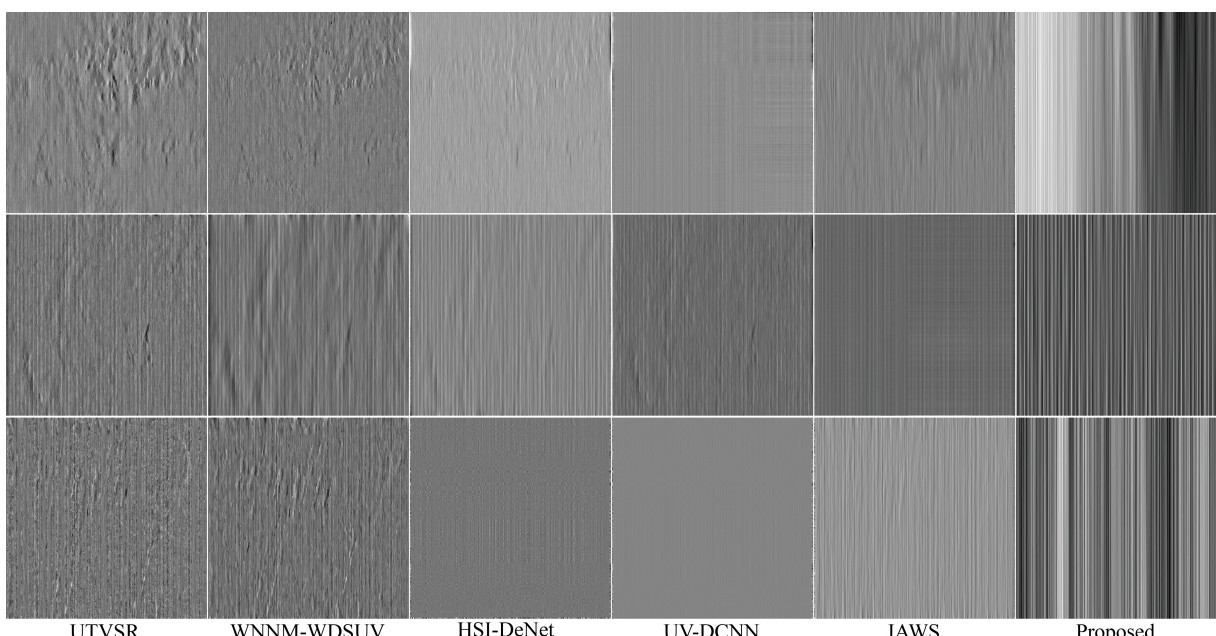

UTVSR　　WNNM-WDSUV　　HSI-DeNet　　UV-DCNN　　JAWS　　Proposed

**Figure 14.** Visual comparisons of the estimated stripes corresponding to the *RAM1*, *RTM1*, and Hyperspectral *urban* images in Figure 13.

**Table 4.** Quantitative comparisons of the state-of-the-arts on the real-life optical RSIs shown in Figure 6.

| Indexes | Methods | *RAM1* | *RAM2* | *RAM3* | *RTM1* | *RTM2* | *RTM3* | *Urban* |
|---------|---------|--------|--------|--------|--------|--------|--------|---------|
| QM | UTVSR | 25.18 | 32.37 | 11.82 | 12.57 | 12.83 | 23.78 | 27.58. |
| | WNNM-WDSUV | 25.47 | 32.69 | 12.24 | 13.08 | 13.26 | 24.33 | 28.81 |
| | HSI-DeNet | 26.39 | 33.97 | 12.89 | 13.67 | 13.81 | 24.85 | 30.49 |
| | UV-DCNN | 26.62 | 34.57 | 13.48 | 14.13 | 14.37 | 25.61 | 31.14 |
| | JAWS | 26.79 | 34.78 | 14.17 | 14.52 | 14.88 | 25.94 | 31.63 |
| | Proposed | **27.11** | **35.27** | **14.72** | **15.18** | **15.49** | **26.37** | **32.16** |
| MICV | UTVSR | 35.72 | 33.26 | 38.19 | 37.54 | 36.47 | 36.79 | 29.46 |
| | WNNM-WDSUV | 35.91 | 33.67 | 38.42 | 37.76 | 36.65 | 36.92 | 29.58 |
| | HSI-DeNet | 36.15 | 33.83 | 38.79 | 37.93 | 36.89 | 37.27 | 29.84 |
| | UV-DCNN | 36.29 | 34.08 | 38.94 | 38.17 | 37.09 | 37.55 | 30.09 |
| | JAWS | 36.67 | 34.41 | 39.18 | 38.54 | 37.51 | 37.86 | 30.57 |
| | Proposed | **36.86** | **34.74** | **39.49** | **38.82** | **37.74** | **38.28** | **31.07** |
| MMRD | UTVSR | 0.45 | 0.67 | 0.051 | 0.058 | 0.062 | 0.36 | 0.57 |
| | WNNM-WDSUV | 0.39 | 0.52 | 0.046 | 0.051 | 0.055 | 0.32 | 0.53 |
| | HSI-DeNet | 0.37 | 0.049 | 0.041 | 0.047 | 0.048 | 0.29 | 0.48 |
| | UV-DCNN | 0.33 | 0.041 | 0.037 | 0.042 | 0.043 | 0.27 | 0.44 |
| | JAWS | 0.29 | 0.036 | 0.031 | 0.036 | 0.038 | 0.22 | 0.35 |
| | Proposed | **0.21** | **0.026** | **0.022** | **0.027** | **0.029** | **0.19** | **0.27** |
| NIQE | UTVSR | 7.03 | 7.37 | 4.18 | 4.22 | 4.26 | 6.73 | 7.19 |
| | WNNM-WDSUV | 6.94 | 7.18 | 4.05 | 4.11 | 4.17 | 6.67 | 7.07 |
| | HSI-DeNet | 6.81 | 7.06 | 3.94 | 4.08 | 4.09 | 6.51 | 6.91 |
| | UV-DCNN | 6.67 | 6.84 | 3.78 | 3.83 | 3.85 | 6.17 | 6.39 |
| | JAWS | 6.46 | 6.61 | 3.62 | 3.67 | 3.71 | 5.95 | 6.08 |
| | Proposed | **6.23** | **6.33** | **3.28** | **3.36** | **3.39** | **5.28** | **5.62** |

For each image, the best result is marked with bold font.

## 5. Conclusions

Random noise (additive Gaussian white noise, AGWN) as well as stripe noise always coexists in remotely sensed images, increasing the difficulty in constructing the inverse problems. To cope with this problem and preserve as more details as possible in estimated remotely sensed image, this paper had proposed a novel dual denoiser driven convolutional neural networks (D$^3$CNNs) which has the following key points: (1) two deep learning networks are trained for different specialized tasks, specially denoising and stripe estimation. (2) The prelearned modules are respectively employed as denoiser and striper priors plugged into the model-based optimization method of HQS to solve the image restoration problem. Experimental results had validated that the two deep powerful priors can improve the effective of model-based methods, with which the proposed D$^3$CNNs strategy yields quite competitive visual and quantitative results compared to the state-of-the-arts. The satisfactory run time is suitable for further applications. Although the proposed method performs well on mixed noise removal (main for two additive noise), it is still to be further extended to reduce other noise, such as impulse noise and Poisson noise.

**Author Contributions:** Z.H.: Investigation, Writing—original draft. Z.Z.: Software. Z.W. and X.L.: Visualization, Investigation. B.X. and Y.Z.: Writing—review and editing. Z.Z.: formal analysis; H.F.: Conceptualization, Methodology. All authors have read and agreed to the published version of the manuscript.

**Funding:** This work was supported in part by the National Natural Science Foundation of China under Grant No. 61901309.

**Data Availability Statement:** Not applicable.

**Acknowledgments:** We appreciate the critical and constructive comments and suggestions from the reviewers that helped improve the quality of this manuscript.

**Conflicts of Interest:** The authors declare no conflict of interest.

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
