# Peer review of "D3CNNs: Dual Denoiser Driven Convolutional Neural Networks for Mixed Noise Removal in Remotely Sensed Images"

_remotesensing, doi:10.3390/rs15020443_

Round 1
Reviewer 1 Report
This paper proposes a novel Dual Denoiser Driven Convolutional Neural Network (D3CNN) to remove random and stripe noise for remotely sensed images. The paper is organized and written well, technical issues are all correct. Experimental results are convincing. However, there are still some major problems.
(1) The contributions of the manuscript are suggested to be enhanced to clearly address the advantages of the proposed method.
(2) How about the convergence of the proposed CNN denoiser-embedded iterative algorithm?
(3) The proposed method is suggested to be summarized in an algorithm environment step-by-step.
(4) If the newly published method (e.g., published in 2022) is added for comparison, the experimental results would be more convincing to show the effectiveness of the proposed method.
(5) Some strongly related mixed noise removal methods should be reviewed and discussed in the Introduction, e.g., 10.3390/rs14051243, 10.1109/TIP.2022.3226406, 10.1109/LGRS.2021.3061541
Author Response
Thank you very much for your comments. My responses are listed in the attachment file, please download and check them. Thank you very much again.

Reviewer 2 Report
Remote sensing images are corrupted by noise, especially mixed noise, causing their quality to be worse, which is not beneficial for us to interpret their contents. To handle the problem of mixed noise removal, the authors reported a denoising scheme, named Dual Denoiser Driven Convolutional Neural Networks (D3CNNs), by joining the merits of model-based optimization methods and discrimination learning approaches. Specifically, D3CNNs have the following critical procedures: (1) D3CNNs denoising model construction with different priors, (2) the model is separated by the alternating direction method of multipliers (ADMM), and (3) the denoising or de-striping parts are replaced by discrimination learning strategies. Finally, both quantitative and qualitative results validate its effectiveness. In my opinion, the paper is easy to be followed, and the theory, methods, and experiments are clear. Therefore, I suggest that it can be published in this journal with the following minor revisions:
1. Eq. (6) should be one line. Similar to Eq. (9).
2. If the momentum batch normalization (MBN) is not proposed by yourselves, please cite the reference.
3. Line 273, “output” should be removed.
4. In Line 377, one author contribution is missing.
Author Response

(The authors gave the same response as above.)

Reviewer 3 Report
This manuscript proposed a novel deep learning-based method for mixed noise removal in remotely sensed images, where a dual denoiser driven convolutional neural network were developed for the task of interest. The proposed network consists of two main parts: 1) two auxiliary variables for denoised image and stripe noise and 2) U-net and RCNN for two variables. Finally, the experiments were conducted to validate the performance of the proposed method, with promissing outcomes. Overall, the topic of this study is interesting, and the manuscript was well organised and written. I suggest that it can be accepted for publication in Remote Sensing, if the authors can well address the following comments.
1. The major contributions and novelty of this study should be clearly highlighted in abstract and introduction.
2. Broaden and update literature review on CNN or deep learning in engineering applications, such as image processing and data analysis. E.g. Vision-based concrete crack detection using a hybrid framework considering noise effect and Magnetorheological Elastomer based torsional vibration isolator for application in a prototype drilling shaft.
3. Generally, the performance of the developed dual denoiser driven CNN is mainly related to the setting of network hyperparameters. How did the authors optimise them in this research to achieve the best prediction performance of trained network?
4. Please illustrate the advantages of the proposed loss function.
5. A parametric study is suggested to evaluate different divisions and network parameters on the model performance.
6. More future research should be included in conclusion part.
Author Response

(The authors gave the same response as above.)

Round 2
Reviewer 2 Report
The paper should be accepted in its current form.